# The Rheology and Textural Properties of Bakery Products Upcycling Brewers’ Spent Grain

**DOI:** 10.3390/foods12193524

**Published:** 2023-09-22

**Authors:** Abirami Ramu Ganesan, Philipp Hoellrigl, Hannah Mayr, Demian Martini Loesch, Noemi Tocci, Elena Venir, Lorenza Conterno

**Affiliations:** 1Division of Food Production and Society, Biomarine Resource Valorisation, Norwegian Institute of Bioeconomy Research, Torggården, Kudalsveien 6, NO-8027 Bodø, Norway; abirami.ganesan@nibio.no; 2Food Technology Area, Institute for Mountain Agriculture and Food Technology Laimburg Research Centre, Laimburg 6, 39051 Pfatten/Vadena, BZ, Italy; philipp.hoellrigl@hotmail.de (P.H.); hannah.mayr@laimburg.it (H.M.); demian.martini-loesch@laimburg.it (D.M.L.); noemi.tocci@laimburg.it (N.T.); elena.venir@laimburg.it (E.V.)

**Keywords:** brewers’ spent grain, texture, rheology, upcycling, techno-functional, side stream, valorization

## Abstract

This study aimed to evaluate the rheological properties of doughs with 50% brewers’ spent grain (BSG) derived from a rye-based (RBSG) and barley-based (BBSG) beer added, and the textural profile of the related baked products. Simple model systems using BSG flour mixed with water were studied. Two bakery products, focaccia and cookies, were made as food systems using BSG in a 1:1 ratio with wheat flour (WF). Their rheological properties and texture after baking were characterized. BSG-added dough exhibited viscoelastic properties with a solid gel-like behavior. The addition of BSG increased G′ > G″ and decreased the dough flexibility. BSG addition in baked RBSG focaccia increased the hardness, gumminess, and chewiness by 10%, 9%, and 12%, respectively. BBSG cookies had a 20% increase in fracturability. A positive correlation was found between the rheological metrics of the dough and the textural parameters of BBSG-added cookies. PCA analysis revealed that complex viscosity, G′, G″, and cohesiveness separated BBSG focaccia from RBSG focaccia and the control. Therefore, the rheological properties of BSG dough will have industrial relevance for 3D-printed customized food products with fiber. Adding RBSG and BBSG to selected foods will increase the up-cycling potential by combining techno-functional properties.

## 1. Introduction

The beer making process also produces a valuable side stream that has not been valorized by producers for decades. This major side stream is brewers’ spent grain (BSG) as well as spent yeast, but to a lesser extent. More than 85% of waste products from the brewing industry are BSG with an estimated 3.4 million tons generated from the European Union [1]. Every 1 kg of beer produces about 0.2 kg of wet BSG. BSG is mainly comprised of polysaccharides, such as cellulose (20–60%), hemicellulose (19–60%), lignin (13–56%), protein (20–25%), and lipids (2.5–6%) [2], whose biochemical composition varies depending on abiotic factors: namely, where the primary raw material grows and the period of harvesting. All these valuable nutrients can be fully utilized as food ingredients; however, because of the 80% moisture content (MC %), BSG must be stabilized. Briefly, preservation can be based on chemical or physical methods. Food-grade preservatives such as propionate-benzoate-sorbate should be used immediately after collection and can extend the shelf life by five days [3]. Organic acids such as formic, lactic, or acetic acid can also be used. However, much longer durability can be achieved by removing water and thus lowering water activity. This is performed using drying techniques such as oven, vacuum, and hot-air drying. Oven drying, vacuum drying, and drum drying have been reported as suitable methods to preserve spent grains, bringing final moisture below 5% [1,4,5].

The valuable nutrients in dried BSG can be used in food formulation, allowing for a circular approach to valorizing waste products. Several researchers have conducted trials using BSG in food such as pasta, bread, cookies, and yogurt. It appeared that BSG increased the water absorption, volume, and texture parameters. The addition of 10% of BSG (total ingredient) improved pasta’s organoleptic and technological properties [6]. On the other hand, fiber supplementation impaired dough processing and product consumer acceptance, owing to the change in rheological properties, loaf volume reduction, increase in crumb hardness, and unsuitable taste and mouthfeel [7,8,9,10]. In general, all food components (fiber, proteins, lipids, and carbohydrates) participate in gel network generation, influencing the viscoelasticity of dough and enhancing the water binding capacity, as extensively reviewed by Wu et al. [11].

Various proportions of BSG in baking products and its influence on textural, sensory, and nutritional outcomes have been explored (Table 1). It was observed that 10 to 20% of BSG (barley-derived) and overall amounts of BSG added do not exceed 30%.

BSG flour addition for baking products was tested and the results showed that having >30% (*w*/*w*) of BSG in the total flour mix reduces the bread’s textural quality [18]. Similarly, cookies with a 3:1 ratio of wheat flour to BSG flour had better taste qualities [19]. Adding 10% BSG flour to crispy slices did not alter the taste and consistency and contributed to the fiber content [20]. Replacing 50% *w*/*w* of the wheat flour with BSG highly exceeded the 20–35% proportion found in the literature, allowing for increased sustainability and valorization potential of BSG.

The main objective of this work was to determine the effect of BSG addition to wheat flour in a 1:1 ratio, on the rheological properties of dough in a simple model system and in two more complex food systems, namely focaccia (flat Italian bread) and cookies, and the resulting effects on the structural properties of baked products. The two complex systems were chosen as examples because they are both baking products widely consumed in the world for their nutrition and convenience [21]. We also wanted to study the BSG behavior in relation to yeast fermentation or to multiple ingredients to evaluate the diverse upcycling possibility of BSG at 50%. Because of the high variation in BSG chemical composition, which depends on malt quality (cereal variety, harvest time, malting condition) and mashing conditions, a 100% barley malt BSG (BBSG) and a 55% rye–45% barley malt BSG (RBSG) were compared. The outcome of this research would also illustrate the function of different BSG flours in selected recipes as guidance for further incorporation in other baking products.

## 2. Materials and Methods

### 2.1. BSG Flour Production

BSG was produced at the Laimburg Research Centre, Italy. A 100% barley BSG (BBSG) was prepared from barley malt and a 55% rye BSG (RBSG) from a rye and barley malt beer-brewing process. BBSG and RBSG moisture contents (MCs) were 68% and 70%, respectively. About 15 kg of each wet BSG underwent a drum drying (Olio CRU s.r.l. Riva del Garda, Italy; patented technology) process to bring down the MCs to 6.2% and 5.4%, respectively. Both dry RBSG and BBSG were subsequently milled to fine flour (Waring blender, Waring commercial, USA). The protein, lipid, carbohydrate, and fiber contents of the BBSG were 16.8 g/100 g, 5.5 g/100 g, 43.3 g/100 g, and 25.3 g/100 g, respectively (Lifeanalytics S.r.l., Oderzo, Italy). The protein, lipid, carbohydrate, and fiber contents of RBSG were 14.2 g/100 g, 4.0 g/100 g, 57.8 g/100 g, and 15.8 g/100 g, respectively (Lifeanalytics S.r.l., Oderzo, Italy). Other physicochemical features of BSG flour were measured. (1) Water activity (a_w_): 0.308 ± 0.011 (BBSG) and 0.352 ± 0.002 (RBSG) measured with Waterlab (Steroglass s.r.l., Perugia, Italy); (2) beta-glucans: 1.29 ± 0.03 g/100 g (BBSG) and 0.95 ± 0.01 (RBSG) measured with a β-Glucan Assay Kit (Megazyme International Ireland Ltd., Wicklow, Ireland); (3) particle sizes: both the BBSG and RBSG flours were mainly composed of particles with length lower than 100 µm; Feret diameter: 91.09% (BBSG) and 83.99% (RBSG); particle size 100–200 µm: 6.53% (BBSG) and 9.06% (RBSG); particle size > 200 µm present at the rates of 2.38% (BBSG) and 6.95% (RBSG); particle size was measured by imaging performed with a Nikon Ti2-Eclipse microscope (Nikon Europe B.V., Amsterdam, The Netherlands) elaborated with Nikon imaging software (NIS Elements AR version 5.30).

### 2.2. Model System

Model systems consisting of BSG, wheat flour (WF), and water were prepared in different proportions described in Table 2. Each dough was prepared in triplicate. In the simple model system, 20 g of WF and BSG flours required 20 and 30 mL of water, respectively, to give doughs with similar consistency/hardness perceived by hand mixing.

### 2.3. Food System Focaccia

Commercial wheat flour, salt, and brewer’s yeast were purchased locally (Supermercato Eurospar, Aspiag Service S.r.l., Auer–BZ–Italy). Standard focaccia dough (WF-F) ingredients (400 g wholemeal wheat spelt flour, 100 g “Manitoba” wheat flour, 12 g salt, 2.5 g brewers’ yeast, and 400 mL water) were mixed at room temperature. In the BBSG and RBSG focaccia (BBSG-F and RBSG-F, respectively) samples, 250 g of the corresponding flour was mixed with 250 g of wholemeal wheat spelt flour, with salt (12 g), brewers’ yeast (2.5 g), and water (400 mL).

All WF-F, BBSG-F, and RBSG-F dough samples were prepared by hand mixing. A damp cloth was used to cover the dough for the 4 h of resting time. Afterward, the dough was pulled and folded a couple of times by hand, covered again for 10 min, and stored at 4 °C overnight. The next day, the dough was left for about 2 h to reach room temperature. Three samples of dough (50 g each) were set aside for the rheological test. The remaining sample was spread on the greased aluminum mold and baked in a preheated oven (RATIONAL CombiMaster Oven, Landsberg am Lech, Germany), at 190 °C with 80% humidity (steam and dry heat mode) for 17 min. After baking, an olive oil–water mix (1:1) was spread on the focaccia before the texture analysis. Each focaccia dough was prepared in triplicate.

Dough moisture and baked product moisture were measured with the aid of a Moisture Analyzer MA 50/1.X2.A Radwag (Radom, Poland).

### 2.4. Food System Cookie

Commercial wheat flour, brown sugar, banana, butter, dark chocolate, baking powder, and salt were purchased locally (Supermercato Eurospar, Aspiag Service S.r.l., Auer–BZ–Italy). The BBSG and RBSG cookies (BBSG-C and RBSG-C, respectively) were prepared from the specific dough using the corresponding BSG flour. Each dough was prepared by hand-mixing the ingredients: BBSG or RBSG flour (90 g), wheat flour (90 g), brown sugar (90 g), baking powder (5 g), mashed banana (100 g), dark melted chocolate (90 g), butter (70 g), and salt (pinch). For the control cookie (WF-C) dough, only wheat flour (180 g) was used and added to brown sugar (90 g), baking powder (5 g), mashed banana (100 g), dark melted chocolate (90 g), butter (70 g), and salt (pinch). The dough was cooled at 4 °C the refrigerator for at least 1 h. Three samples (50 g each) were taken for the rheological test. After cooling, 20 g portions were flattened into equal-dimensioned pieces and placed in a baking tray with parchment paper. These cookie doughs were oven-baked (RATIONAL CombiMaster Oven, Landsberg am Lech, Germany) at 165 °C for about 20 min. The cookies were cooled to room temperature before the texture analyses. Each cookie dough was prepared in triplicate. The dough moisture and baked product moisture were measured with the aid of a Moisture Analyzer MA 50/1.X2.A Radwag (Radom, Poland).

### 2.5. Rheological Tests for Dough

The measurements were performed at 25 °C in the small oscillation range using an MCR 102 Anton-Paar rheometer (Anton Paar GmbH, Graz, Austria), in a plate–plate geometry system with a diameter of 50 mm and a test gap of 2 mm. Amplitude tests were performed in triplicate at 1 Hz at increasing shear stress to determine both the linear viscoelastic region (LVER) with the related yield point, and the cross-over point (COP), the latter also representing the flow point. 

Mechanical spectra were obtained at shear stress within the LVER in the frequency range from 0.1 HZ to 10 Hz. The elastic modulus (G′), viscous modulus (G″), complex modulus (G*), loss tangent (tan delta), complex viscosity (η*), phase shift angle, and shear strain% were compared at 1 Hz. Three measurements were obtained from each dough replicate.

### 2.6. Texture Profile Analysis (TPA)

The hardness, fracturability, adhesiveness, springiness, cohesiveness, gumminess, chewiness, and resilience parameters were evaluated for the focaccia and cookies using the textural profile analyzer (TA.XT plus texture analyzer, Stable Micro Systems Ltd., Godalming, UK). These properties mimic the TPA mastication process using double compression cycles. A 25 kg load cell (500 N), a 45 mm cylindrical probe for focaccia, and a 36 mm cylindrical compression probe for cookies were applied at a speed of 5.0 mm/s and a trigger force of 5 g. Compression was performed up to a displacement equal to 75% of the sample height. The test parameters were as follows: 10.0 mm/s pre-test speed, 5.0 mm/s test speed, 5.0 mm/s post-test speed, and 40% strain. Three square portions of focaccia and three equal dimensions/sizes of cookies were measured for each dough replicate. All these tests were performed at room temperature, after baking. Measurements of equal-sized portions were taken in triplicate.

### 2.7. Statistical Analysis 

One-way ANOVA statistical analysis and Tukey post hoc test at a significance level α ≤ 0.05, and Pearson correlation analysis at significance levels of 0.05 and 0.01, were carried out using Minitab version 19.2.0 (State College, PA, USA). Principal component analysis (PCA) was carried out using MetaboAnalyst 5.0 [22]. 

## 3. Results and Discussions

### 3.1. The Rheological Properties of the Model Systems

Five simple dough models made with (1) pure wheat flour (WF), (2) pure RBSG, (3) pure BBSG or (4) RBSG, and (5) BBSG in a 1:1 ratio with WF (RBSG50 and BBSG50, respectively) were characterized for defined rheological properties. The WF model was tested in two variants with different WF:water ratios: (a) 20:30 to have the same flour:water ratio as in the model prepared with the BSG flour, and (b) 20:20 to reach a consistency similar to the model dough prepared with BSG.

#### 3.1.1. Amplitude Sweep

Amplitude sweep assays were carried out to determine the linear viscoelastic region (LVER). G′ and G″ moduli versus shear stress were almost parallel within the LVER (Appendix A) with G′ > G″ indicating a gel-like solid structure. All the samples measured in the study could be defined as viscoelastic solid material. After LVER, G′ slowly decreased in all the samples, therefore indicating the gradual breakdown of the network, up to the cross-over point (COP). At the COP, with G′ = G″, the viscous behavior prevailed, and after the COP, where G″ > G′, the material started to flow. The amplitude assay allowed us to establish the value of the yield point or yield stress and the shear strain % at the COP, also called flow point or flow stress. The shear stress at the COP has been indicated as the force required to break the bonds of the gel network [23].

The shear stress and shear strain at the COP are described in Figure 1. The dough obtained from pure WF with 30 g of water (WF 20:30) exhibited lower shear stress, compared to the dough prepared with less water (WF 20:20), and behavior similar to the BSG-added doughs. In fact, the WF 20:30 appearance could be defined as slurry. Nocente et al. [6] reported the requirement for different water amounts in semolina and semolina/BSG blends for optimal dough development and a similar magnitude of enrichment was seen in our study.

Nevertheless, pure BSG doughs showed significantly (*p* ≤ 0.05) higher shear stress at the COP (467.52 ± 72.41 Pa for RBSG and 157.03 ± 22.35 Pa for BBSG), whereas shear stress was much lower in the 50% BSG-added model doughs (26.22 ± 2.19 Pa for RBSG50 and 16.84 ± 6.25 Pa for BBSG50). RBSG had higher shear stress compared to all other doughs. It can be speculated that the low amount of gluten and the high polysaccharide content (maltodextrin, cellulose, and hemicellulose) influenced shear stress and increased the hardness property of the RBSG dough. Among starches, other biopolymers appeared to influence the dough viscosity. For instance, carboxymethyl cellulose increases viscosity and positively affects the extensibility, elasticity, and structure of dough [24].

All BSG model doughs exhibited similar shear strain %, which were lower than in both the WF 20:20 and the WF 20:30 doughs, despite the different water ratios (Figure 1). WF 20:20 was the most flexible dough, with a shear strain at COP of 16.13 ± 3.16%, followed by WF 20:30 (12.60 ± 3.35%). According to Öhrlund et al. [25], the shear strain can be considered an index of flexibility.

BSG addition to model doughs reduced the deformation percentage at the flowing point, regardless of BSG content (100% or 50%); for instance, BBSG and RBSG had a deformation potential (2.85% and 3.29%, respectively) lower than both the WF 20:30 and WF 20:20 models (12.6% and 16.13%). 

Mixing BSG and WF acted on two fronts: it shifted the shear stress values at the flow point of pure BSG dough downwards, i.e., towards the values of pure WF dough, while simultaneously reducing the shear strain values, hence also the deformation of WF dough to values similar to those of pure BSG dough. Gluten is certainly involved in the elastic properties of doughs and the addition of BSG impaired the network deformability, probably because of a lower number of gluten linkages in BSG-added doughs, leading to low extensibility at the COP. It was reported that the addition of bran (fiber) influenced trans-conformational changes in gluten and created the interaction of gluten/water in the dough [26]. This transformation leads to the deformation of the microstructure between gluten linkages, mainly due to the hygroscopic nature of bran. This outcome was also observed in other studies. For instance, rye flour (10 to 20%) mixed with barely BSG increases the absorption of water by 4 to 6%. This shows that BSG (barley- or rye-derived) had the tendency to absorb more water due to the presence of high fiber compared to WF [27].

#### 3.1.2. Frequency Sweep

Mechanical spectra were determined for all samples within the LVER. Reference plots are reported in the Appendix A. As expected from amplitude tests, all samples exhibited G′ > G″ overall frequency ranges and phase shift angles set below 30 degrees, indicating viscoelastic gel behavior. The dynamic moduli G′ and G″ showed an increasing trend with increasing frequency indicating that, in all samples, a strong intermolecular cross-linking effect occurred [28]. 

The G* observed for the dough described in this study is shown in Figure 2. In the LVER, RBSG50 and BBSG50 exhibited stiffness (G*) in between pure BSG and pure WF, although the deformation and phase shift angle were more like those observed in WF doughs. The phase shift angles of BSG-added doughs were not statistically different from pure WF dough. G* has been associated with the stiffness of the material, i.e., the resistance against deformation [23]. All BSG-based doughs showed the highest G* compared to all other samples. 

The elastic characteristics can be attributed to the gluten network, which was able to support shear stress with a significant proportion of BSG. The addition of fiber-rich components (cellulose, etc.) increased the dough consistency and thus its resistance to deformation, presumably due to a higher hydration and water-holding capacity (WHC), together with a decrease in flexibility. Indeed, the resistance to extension and dough extensibility are dependent on the variety and quality of the gluten. Dokić and coworkers [29] found that a 20% chestnut flour addition, owing to lower gluten content, decreased dough extensibility. A similar effect was observed in our study after adding BSG flour. As the gluten network is responsible for elasticity, the hardness (G*) could be strongly affected by the cellulose content of BSG. 

It was observed that β-glucan fragments’ β-(1→4) bonds possessed larger intermolecular connections, resulting in stronger bonds between dextran chains (barley-based BSG), which results in greater flow resistance and high viscosity in the dough. On the other hand, cellulose tetrasaccharide-based units and β-(1→4) bonds from beta-glucan have an important role in increasing the viscosity and gel formation [30]. 

The WF dough resulted in much lower complex moduli; nevertheless, it gave a similar strain %, with deformation at the COP up to 15% (Figure 1). BSG-added doughs differed in two characteristics. They were harder than the WF dough since they exhibited higher complex moduli, and this could be attributed to the high fiber content of BSG which absorbs more water, hence increasing WHC [31], compared to the wheat flour. In addition, all BSG-added doughs were less deformable, probably because of the lower content of gluten (glutenins and gliadins) responsible for the extensive network able to confer elasticity to the dough [31].

### 3.2. The Rheological Properties of the Food Systems

BSG flour was used as a food ingredient for the development of recipes substituting 50% of the original wheat flower ingredient with BSG flour. Preliminary tests were carried out to verify the feasibility of this addition (data not shown). According to the bibliometric analysis, a maximum amount of BSG flour of 35% (dry basis) has been reported for mixing with standard flour for bakery products (crispy bread, muffins, baked snacks, etc.) [32].

#### 3.2.1. Focaccia Food Model

The WF-F, BBSG-F, and RBSG- F dough moisture contents were 43.0 ± 0.3% *w*/*w*, 47.8 ± 2.6% *w*/*w*, and 37.1 ± 8.0% *w*/*w*, respectively. 

The moisture contents of the WF-F, BBSG-F, and RBSG- F baked products were recorded as 41.6 ± 2.1% *w*/*w*, 39.7 ± 1.0% *w*/*w*, and 39.6 ± 0.5% *w*/*w*, respectively.

Moisture content was not significantly different either among the baked products or among the doughs, meaning that the detected differences in rheological and texture parameters would be ascribable to the solid part composition and not to the different moisture contents.

The shear stress and strain % of focaccia were measured at the end of the LVER (yield point) and at the COP (flow point) (Table 3). 

The shear stress at both the yield point and the flow point was significantly lower in the case of pure WF-F dough when compared to both RBSG-F and BBSG-F doughs. The addition of BSG flour reduced the strain % at the COP slightly but significantly, which can be related to the flexibility of the focaccia doughs.

The dynamic moduli and complex viscosity of samples were compared at a frequency of 1 Hz. In Figure 3, it can be observed that BSG-added focaccia doughs exhibited higher G′, G″, and G* moduli and complex viscosity compared to the control, regardless of the barley malt percentage in the BSG; no statistical differences were determined between RBSG-F- and BBSG-F-added doughs.

It could be inferred that WF-F partial substitution with 50% BSG could change the malleability of products involving elastic properties in baking. Peressini and Sensidoni [10] investigated the effect of the addition of dietary fiber (inulin at different degrees of polymerization) on the rheological and breadmaking properties of wheat doughs. They reported that inulin with a high degree of polymerization highly influences the viscoelastic properties of dough, contributing to the strength of the dough, and increasing elasticity. In the same study, the increase in solid-like behavior was inversely related to the dough expansion during fermentation. It was also observed that the fiber can directly influence the dough elasticity, while indirectly affecting the variation in the water–flour ratio (absorption). Dynamic moduli (G′, G″, and G*) are very sensitive to water content; these moduli increase while water content decreases [10]. It was reported that partial and full hydration induced the quick formation of a more compact gluten network when compared to limited hydration [33]. Similarly, we can hypothesize a direct and indirect effect of BSG addition: it directly affects the gluten network and, therefore, the elastic nature of doughs, reducing its elasticity, and indirectly it affects the WHC, increasing the hardness of dough by limiting the water availability for plasticization. The cellulose network, on the other hand, is responsible for higher WHC, thus reducing its plasticizing availability. It was reported that unmilled cellulose had a low WHC (3.44 mL/g) compared to milled cellulose (14.03–24.99 mL/g), and this property is also related to the bulk density, swelling capacity, crystallinity, and surface area of the food system [31]. In our study, the cellulose in the milled BSG flour might be responsible for the WHC and crystallinity with a behavior comparable to the outcome reported by Dubey et al. [31]. 

#### 3.2.2. Cookie Food Model

The WF-C, BBSG-C, and RBSG-C dough moisture contents were 14.0 ± 0.6% *w*/*w*, 11.8 ± 0.4% *w*/*w*, and 13.3 ± 0.5% *w*/*w*, respectively. The moisture contents of the WF-C, BBSG-C, and RBSG-C baked cookies stood at 7.9 ± 0.6% *w*/*w*, 6.0 ± 1.1% *w*/*w*, and 5.7 ± 0.5% *w*/*w*, respectively. The BBSG-added dough moisture content was significantly lower than those of both the control and the RBSG-added dough. After baking, both the BBSG- and the RBSG-added cookies showed a significantly lower moisture content compared to the control. However, since in all samples the moisture content is much less than 40%, it can be assumed that there was no formation of a gluten network [34].

The cookies were prepared with a greater variety and amount of ingredients, resulting in the BSG flour addition having less influence on the dough properties. BSG addition gave a significant (*p* < 0.05) increase in the shear stress at COP (297 ± 0.00 Pa) and a slight decrease in shear strain% (17.3 ± 1.41%) in comparison to the control dough (193.3 ± 17.4 Pa and 20.7 ± 3.3%, respectively). Hence BSG-added dough required higher shear stress to deform, resulting in lower flexibility compared to pure WF dough. 

Figure 4 shows the dynamic moduli and complex viscosity measured at a frequency of 1 Hz in cookie dough. G′, G″, G*, and complex viscosity measured in BBSG-C were significantly higher than in RBSG-C and WF-C. This could be due to the presence of a high amount of hemicellulose and cellulose in BBSG–55.07 g/100 g (dry matter basis) compared to rye consisting of 9.92 g/100 g (55% rye in RBSG) [27].

Two main components responsible for the solid-like behavior of doughs in the present study can be highlighted: the gluten network and the fiber content. Even though starch might play a role in the dough, this should not apply to BSG flour since most of it is expected to hydrolyze during the mashing process. Furthermore, the residual quantity of starch BSG addition should have a negligible influence when compared to the starch contribution of the wheat flour. The percentage of water in the dough will determine which of those components is mostly involved in the solid-like structure. Indeed, the gluten network does not form if the water content is below 40%; above 40% an extensive network can form; and a further increase in water up to 50% does not modify the dough structure but only has a plasticizing effect [34]. Thus, the gluten network is responsible for the dough structure in samples containing at least 40% water, but it becomes much less relevant when water is below this value. 

In the cookie dough prepared in our study, where the only water was from the banana ingredient, and the moisture content was below 14%, the gluten network did not form. In addition, in the cookie dough, the difference in cellulose and other hydrocolloid content between RBSG and BBSG accounted for the different hardnesses (dynamic moduli) if compared to the simple model systems or focaccia samples. In simple model systems, the percentage of water was between 50 and 60%; therefore, in the more than 40% group discussed by Berland and Launary [34], the gluten network formed properly and gave elasticity and hardness. Increasing the water content in the dough (WF 20:30) decreased the hardness compared to WF 20:20. Similarly, focaccia dough consisting of equal proportions of water and flour in BBSG-F and RBSG-F recorded stronger hardness than WF-F dough, with no significant difference between the RBSG and BBSG focaccia doughs. In the case of cookies, no water or watery solutions were added, preventing the formation of a gluten network, and this probably gave the BSG fibers (both qualitative and quantitative) a predominant role in determining hardness. Therefore, the type of fiber and its quantity plays a preponderant role in cookies compared to the gluten network. BBSG exhibited a much higher hardening capacity than RBSG because of the different content in cellulose.

BSG significantly changed the storage modulus, a property correlated with the proofing and final volume of baked products [35]. As previously observed by Ktenioudaki et al. [35], after BSG addition (15 to 35%) the storage and loss moduli of the doughs increased, indicating a more solid-like behavior with a low flow point which negatively affected the rheological properties such as biaxial extensional viscosity and uniaxial extensibility of doughs. In addition, β-glucan content was reported to influence the rheology of barley flour dough [36], indicating that this compound could also be responsible for the significant difference in the BBSG cookies compared to the RBSG-C and WF-C. It can be suggested that identifying β-glucan in the BSG might be helpful in designing an appropriate mixture in the future.

### 3.3. Texture Profile Analysis (TPA) 

Texture profile analysis was carried out to identify the motion of food under instrumental conditions to mimic the human jaw movement. Hardness, springiness, chewiness, and gumminess were evaluated in the baked focaccia and cookies, and the fracturability of the cookies was measured. Each product was measured after baking on the same day. Three replicates in three batches were used for each test. 

#### 3.3.1. Focaccia Food Model

The springiness was similar in all focaccia types, while the hardness, gumminess, cohesiveness, and chewiness showed significant differences among samples (Table 4). Higher hardness, gumminess, and chewiness were recorded for RBSG-F compared to WF-F and BBSG-F. These results make the product of this study comparable to other rye-incorporated bread showing higher chewiness and hardness [37] due to the presence of fiber. Chewiness, related to the capacity to change the food structure from chewable to swallowable, was significantly different between BBSG focaccia and WF. This could be due to changes in the structure of arabinoxylan, cellulose, and hemicellulose which lead to water loss, cell wall integrity, and turgor pressure during baking [38]. 

The gumminess and chewiness of RBSG samples were significantly higher (*p* < 0.05) compared to the other two samples. Other focaccia made from liquid sourdough (1.5 g/100 g) had a hardness (maximum peak force) of 8679.6 g, which is higher than the RBSG of the present study (26.36 N, equivalent to 2688.72 g). In other studies, 5% BSG-added muffins showed almost similar hardness of 26.14 N [39], and 10.91 N was reported in 5% BSG bread [40]. It was observed in bread that lower hardness lowers the chewiness and gumminess [41] these results are on par with the present study. BSG addition in sourdough bread affects the crumb hardness, and also increases the dough development time from 1.27 to 6.16 min and from 3.37 to 5.42 min [42]. In addition, 5 to 15% BSG muffins lowered the gumminess (8.07 to 29.76 N) and chewiness (7.44 to 28.29 N) properties [39]; however, in our study, different trends were observed in the texture, i.e., RBSG increases gumminess (11–19.12 N) and chewiness (10–24.98 N) compared to the control WF. This difference can be due to the fact that enzymatic treatment was applied to the BSG muffins whereas the present study used raw BSG flour without any treatment.

#### 3.3.2. Cookie Food Model

Higher hardness, chewiness, and gumminess together with lower fracturability were measured for the RBSG cookies. It was also noted that no significant differences were found in springiness and cohesiveness (Table 5). 

A higher hardness of 57.2 to 90.59 N was reported for the BSG cookies [43] compared to what was observed in our study on BBSG cookies; nevertheless, different drying treatments were applied to the BSG, such as microwave and conventional oven drying, whereas in the case in the present study, drum drying was applied. By contrast, a decrease in hardness was observed in 20% BSG-added biscuits with respect to wheat flour biscuits from ~1100 g to 1000 g [44]. BSG addition acted positively on sensory qualities such as hardness and crispiness for 20% BSG cookies baked at 200 °C for 12 min [45]. Based on these data, the BSG-added cookies are expected to give better quality scores in terms of sensory qualities, and further work must be carried out to ascertain this hypothesis. Overall, BSG addition to cookies showed a variable effect on the hardness depending on the BSG percentage used [12]. Barley origin, harvesting time, malting, and mashing process have also been reported to be responsible for the difference in BSG chemical compositions [46]. 

### 3.4. Rheological and Textural Parameters Correlation

The correlation between the rheological properties of focaccia dough and the textural parameters of the related baked product was evaluated using the Pearson coefficient. No significant correlations were found between hardness, gumminess, chewiness, and G′ with G″, G*, and complex viscosity measured for focaccia (Table 6).

Regarding cookies (Table 7), the rheological dynamic moduli G′, G″, and G*, and complex viscosity showed high significant positive correlations with hardness and gumminess (r = 0.76 to 0.97, *p* > 0.01). A highly significant correlation was found also between gumminess and resilience influenced by G′, G″, G*, and complex viscosity (r = 0.76 to 0.9, *p* > 0.01). Waters et al. [47] found in 20% BSG-added bread a high correlation between G* modulus and hardness, and springiness. BSG-added muffins (15%) showed a positive correlation with the hardness and the gumminess [37], being on par with the present results. Resilience is the property of regaining the original shape after the removal of deformation force [48], and it was significantly influenced by G′, G″, G*, and complex viscosity in the cookies. 

#### Principal Components Analysis (PCA)

In order to analyze the comprehensive features arising from the rheology and texture profile of the complex food models “cookie” and “focaccia”, a principal component analysis (PCA) was carried out. The results of the PCA are represented for focaccia (Figure 5a,b) and cookies (Figure 6a,b).

In focaccia, the PCA score plot and biplot (Figure 5a,b, respectively) showed the first two principal components accounting for 81.3% of the total variance (PC1, 62.8%, and PC2, 18.5%). Samples were mainly separated based on PC1, with WF clustering on the left side of the plot and RBSG and BBSG mainly on the right side, showing the higher similarity of the BSG-added product. Textural parameters such as springiness, hardness, and gumminess contributed to the separation of RBSG focaccia from the other focaccia samples. Shear strain% and phase shift angle were negatively correlated with the storage modulus, and the loss modulus (G′, G″), appeared in the opposite quadrant of PC1, driving the separation between WF and BBSG focaccia samples.

WF, RBSG, and BBSG exhibited dissimilarities in the score plot (Figure 6a) based on texture and rheology variables, as in the biplot (Figure 6b). PC1 (43.7%) and PC2 (25.9%), accounted for 69.6% of the total variance. The hardness, gumminess, chewiness, cohesiveness, and resilience parameters cluster together, representing RBSG samples. Phase shift angle and shear strain% were in the negative quadrant in which WF mainly cluster. PC1 allowed the separation of BBSG cookies under the influence of complex viscosity, G moduli, and fracturability, placing the samples on the right side of the plot.

## 4. Conclusions

With the emergence of the circular economic approach in recent times, it is profitable, sustainable, and ethical to fully utilize BSG in the food supply chain; therefore, this study utilized 50% DM in the food mix, emphasizing a higher proportion of BSG inclusion in selected baked foods. The effect of BSG addition was studied in both simple model systems (doughs) and food products (doughs and baked products).

Both BBSG and RBSG addition to doughs affects the rheological properties of both simple models and food systems, also depending on the proportion of water. All doughs exhibited rheological properties typical of viscoelastic materials, with a solid-gel-like behavior. The interplay between the microstructure of fibers increased the viscoelasticity (tanδ) and improved the hardness of BSG-added products. Moreover, G′ and G″ showed increasing trends with the addition of rye in BSG, i.e., RBSG compared to BBSG (G′ > G″ in rye and barley concentrations), thereby leading to an enhancement in the network structure. 

The addition of BSG increases hardness and decreases deformability, probably owing to a weaker gluten network and more fiber contributing to higher water-holding capacity. This fiber can be arabinoxylan, cellulose, hemicellulose, β-glucan, and dextrin in BSG, but depends on the different degrees of polymerization between RBSG and BBSG.

RBSG addition enhances the overall textural profile for focaccia, and BSG enhances the fracturability of cookies. Therefore, upcycling the rye beer brewing side-streams in small breweries and craft beer in combination with barley will enhance the nutritional value and the textural acceptability of final products.

Studies on the rheology of BSG dough (simple model system) and its influence on the textural properties (food system) of high-proportion BSG (50%) products are limited. At the industrial scale, upcycling BSG up to 50% will significantly reduce the addition of imported wheat flour and associated product costs. This will eventually increase the functionality of BSG along with fiber in suitable foods. Knowing the rheological behavior of BSG dough and the related textural properties of baked products will help the industry to produce customized food designs using BSG with fiber. In future perspectives, this research might give a better insight into 3D food printing techniques with precisely controlled use of BSG in product development.

## Figures and Tables

**Figure 1 foods-12-03524-f001:**
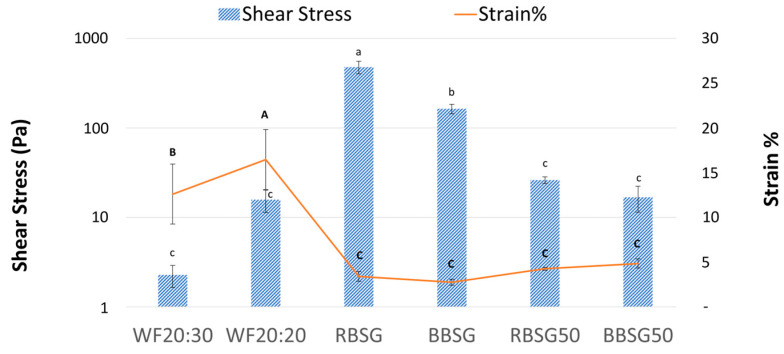
Shear stress (Pa) and shear strain % at the cross-over point measured in simple model doughs. Blue dashed bars: shear stress; orange broken line: shear strain; WF 20:30 (=20 g wheat flour + 30 g water) WF 20:20 (=20 g wheat flour + 30 g water); RBSG (=20 g rye-based BSG flour + 30 g water); BBSG (=20 g barely-based BSG flour + 30 g water); RBSG50 (=10 g wheat flour + 10 g rye BSG flour + 30 g water); BBSG50 (=10 g wheat flour + 10 g barley BSG flour + 30 g water). Average value of at least three replicates; error bars represent the standard deviation; different lowercase letters denote statistically significant shear stress values (*p* ≤ 0.05); different capital bold letters denote statistically significant shear strain values (*p* ≤ 0.05).

**Figure 2 foods-12-03524-f002:**
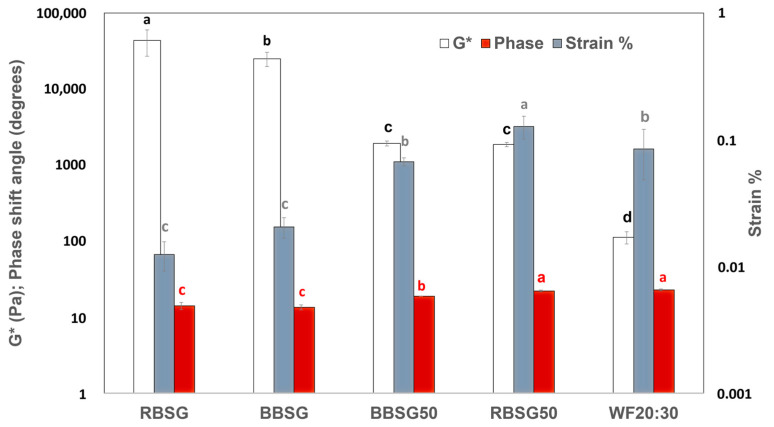
G* (complex moduli), phase shift angles, and strain % at 1 Hz frequency of the simple model system. RBSG—raw Rye + barley BSG flour 20 g in 30 mL water; BBSG—raw barley-derived BSG flour 20 g in 30 mL water; RBSG50—wheat flour 50% + rye BSG flour 50% 20 g in 30 mL water; BBSG50—wheat flour 50% + barely BSG flour 50% 20 g in 30 mL water. WF 20:30 = wheat flour 20 g in 30 mL water. Different lowercase letters denote statistically significant different values, within the same variable (black lowercase letter: G*; red lowercase letter: phase shift angles; grey lowercase letter: strain %).

**Figure 3 foods-12-03524-f003:**
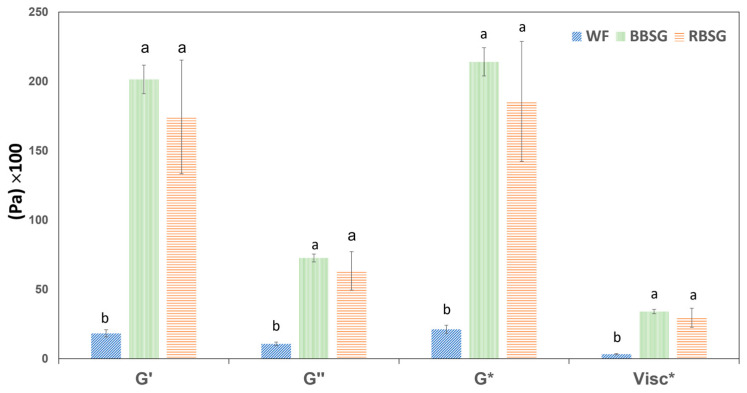
Elastic (G′), viscous (G″), and complex (G*) moduli, together with complex viscosity (Visc*) at a frequency of 1 Hz of focaccia doughs obtained with or without the addition of BSG flour. Mean ± standard deviation of three replicates; different lowercase letters denote significantly different means (*p* ≤ 0.05).

**Figure 4 foods-12-03524-f004:**
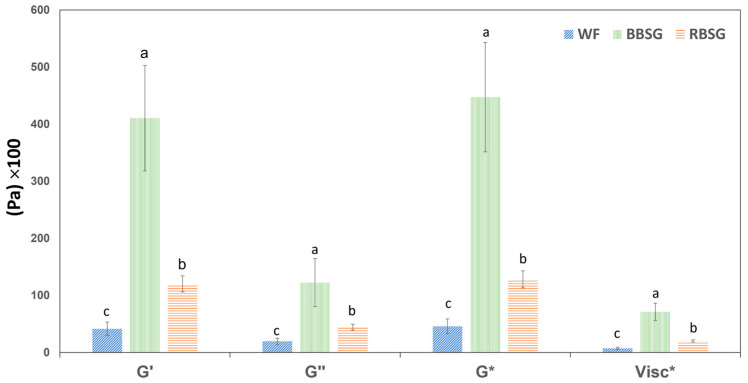
Elastic (G′), viscous (G″), and complex (G*) moduli, together with complex viscosity (Visc*) at a frequency of 1 Hz of cookie doughs obtained with or without the addition of BSG flour. Mean ± standard deviation of three replicates; different lowercase letters denote significantly different means (*p* ≤ 0.05).

**Figure 5 foods-12-03524-f005:**
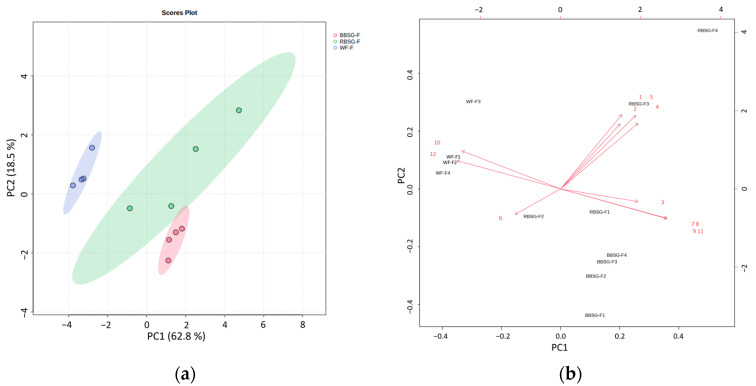
Principal component analysis of the rheology and texture measurements taken for the focaccia model recipe, with PC1 accounting for 62.8% of the variance and PC2 accounting for 18.5% of the variance. (**a**) Score plot GREEN dots and area: RBSG-F (rye-based BSG:wheat flour—1:1 ratio), RED dots and area: BBSG-F (barley-based BSG:wheat flour—1:1 ratio), and BLUE dots and area WF-F (only wheat-flour-based recipe used as control). (**b**) Loading plot of the variables (red arrows). RBSG-F (rye-based BSG:wheat flour—1:1 ratio), BBSG-F (barley-based BSG:wheat flour—1:1 ratio), and WF-F (only wheat-flour-based recipe used as control). (1) Hardness; (2) springiness; (3) cohesiveness; (4) gumminess; (5) chewiness; (6) resilience; (7) G′; (8) G″; (9) G*; (10) phase shift angle; (11) complex viscosity; (12) shear strain %.

**Figure 6 foods-12-03524-f006:**
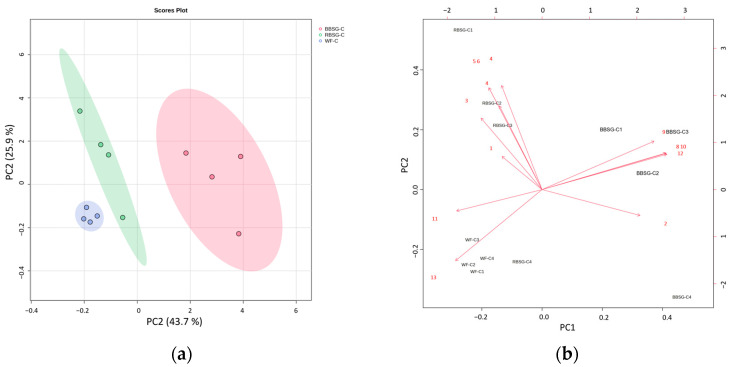
Principal component analysis of the rheology and texture measurements taken for the cookie model recipe, with PC1 accounting for 43.7% of the variance and PC2 accounting for 25.9% of the variance. (**a**) Score plot GREEN dots and area: RBSG (rye-based BSG:wheat flour—1:1 ratio), RED dots and area BBSG (barley-based BSG:wheat flour—1:1 ratio), and BLUE dots and area WF (only wheat-flour-based recipe used as control). (**b**) Loading plot of the variables (red arrows) RBSG (rye-based BSG:wheat flour—1:1 ratio), BBSG (barley-based BSG:wheat flour—1:1 ratio), and WF (only wheat-flour-based recipe used as control). Red numbers indicate the variables: (1) hardness; (2) fracturability; (3) springiness; (4) cohesiveness; (5) gumminess; (6) chewiness; (7) resilience; (8) G′; (9) G″; (10) G*; (11) phase shift angle; (12) complex viscosity; (13) shear strain %.

**Table 1 foods-12-03524-t001:** BSG-added baking products as a percentage of the total flour and its influences on the product rheology and texture. Summary of the major findings from the literature published in the past 5 years.

Products	% BSG Addition	Main Outcome
Cookies [12]	10%, 20%, and 30% BSG in cookies with wheat flour	BSG (30%) increases dough development time with lower sensory qualities.
Biscuits [13]	10% BSG in wheat and oat flour mix (baking powder)	BSG + oat resulted in greater consumer acceptance. Influences textural properties like chewiness, stickiness, being pastry-like, and hard to swallow in dynamic sensory perception.
Bread [14]	20% BSG with corn grits; sugar beet pulp; apple pomace	BSG as a coextrude increases the dietary fiber and lowers the sensory acceptability.
Biscuits [15]	15% and 30% BSG with wheat flour	Textural properties were not appealing in 30% BSG after *Rhizopus oligosporus* metabolic changes.
Bread, breadsticks, and pizza [16]	5% and 10% BSG with soft wheat flour	BSG (10%)-added dough had higher water absorption and lower dough development time. BSG (5%)-added baked products had higher stability, strength, and tenacity.
Bread [17]	20% and 25% of BSG, BSG-gluten with Manitoba soft wheat flour	BSG-added bread increased resistance to chewing and the fibrousness decreased the crust crispness.

**Table 2 foods-12-03524-t002:** Barley BSG (BBSG), rye BSG (RBSG), wheat flour (WF), and water ratios used to prepare the simple model systems (**a**) and complex food systems (**b**).

(a) Simple Model Systems	Sample Code	Dry Flour (g)	Water (g)
Wheat flour	WF 20:20	20	20
Wheat flour	WF 20:30	20	30
BBSG flour	BBSG	20	30
RBSG flour	RBSG	20	30
50% BBSG + 50% Wheat flour	BBSG50	10 + 10	30
50% RBSG + 50% Wheat flour	RBSG50	10 + 10	30
**(b) Food Systems**	**Sample Code**	**Dry Flour Ratio**	**Other Ingredients**
Focaccia (Control)	WF-F	WF 100%	See Section 2.3
BBSG focaccia	BBSG-F	WF50% + BBSG50%	See Section 2.3
RBSG focaccia	RBSG-F	WF50% + RBSG50%	See Section 2.3
Cookies (Control)	WF-C	WF 100%	See Section 2.4
BBSG cookies	BBSG-C	WF50% + BBSG50%	See Section 2.4
RBSG cookies	RBSG-C	WF50% + RBSG50%	See Section 2.4

**Table 3 foods-12-03524-t003:** Shear stress and shear strain values at the linear viscoelastic region (LVER) and COP of focaccia (F) doughs prepared with pure wheat flour (WF), RBSG (rye-based BSG) and wheat flour (1:1), or BBSG (barley-based BSG) and wheat flour (1:1).

Samples	LVER		COP	
	Yield Point Shear Stress (Pa)	Strain %	Flow Point Shear Stress (Pa)	Strain %
WF-F	1.6 ± 0.24 ^b^	0.07576 ± 0.0189	70.6 ± 17.3 ^b^	24 ± 4.64 ^a^
BBSG-F	14.3 ± 4.45 ^a^	0.0938 ± 0.0174	380.1 ± 142.6 ^a^	16.3 ± 2.62 ^b^
RBSG-F	13.3 ± 2.12 ^a^	0.0731 ± 0.0185	375.8 ± 58.8 ^a^	9.3 ± 2.25 ^c^

LVE—linear viscoelastic range; COP—cross-over point; mean ± standard deviation of three replicates; different lowercase letters denote significantly different means (*p* ≤ 0.05). The absence of a superscript letter denotes no significant difference.

**Table 4 foods-12-03524-t004:** Textural parameters of focaccia with or without the addition of BSG.

Focaccia	Hardness (N)	Springiness	Gumminess (N)	Chewiness (N)	Resilience	Cohesiveness
WF-F	16.46 ± 6.25 ^b^	1.134 ± 0.3	10.06 ± 0.12 ^b^	12.22 ± 4.06 ^b^	0.4669 ± 0.04	0.6218 ± 0.01 ^b^
RBSG-F	26.36 ± 8.05 ^a^	1.253 ± 0.2	19.12 ± 0.16 ^a^	24.98 ± 8.18 ^a^	0.46417 ± 0.02	0.7181 ± 0.02 ^a^
BBSG-F	14.7 ± 3.52 ^b^	1.082 ± 0.3	11.59 ± 0.09 ^b^	10.78 ± 3.13 ^b^	0.46126 ± 0.02	0.7878 ± 0.02 ^a^

Different superscript letters within each column denote significant differences (*p* < 0.05) between samples. The absence of a superscript letter denotes no significant difference. Data are expressed as mean ± SD.

**Table 5 foods-12-03524-t005:** Textural parameters of cookies with or without the addition of BSG.

Cookies	Hardness (N)	Fracturability (N)	Springiness	Cohesiveness	Gumminess (N)	Chewiness (N)	Resilience
WF-C	3.54 ± 1.9 ^b^	34.91 ± 22.7 ^ab^	0.938 ± 0.1	0.166 ± 0.07	0.654 ± 0.6 ^b^	0.654 ± 0.7 ^b^	0.040 ± 0.0
RBSG-C	24.33 ± 13.5 ^a^	21.18 ± 6.25 ^b^	1.188 ± 1.0	0.149 ± 0.06	3.324 ± 2.8 ^a^	3.633 ± 3.6 ^a^	0.047 ± 0.0
BBSG-C	1.897 ± 0.4 ^b^	54.25 ± 27.9 ^a^	0.84 ± 0.1	0.122 ± 0.07	0.239 ± 0.1 ^b^	0.214 ± 0.1 ^b^	0.037 ± 0.0

Different superscript letters within each column denote significant differences (*p* < 0.05) between samples. The absence of a superscript letter denotes no significant difference. Data are expressed as mean ± SD.

**Table 6 foods-12-03524-t006:** Pearson correlations between rheological and textural parameters of focaccia. Data from WF and BSG-added focaccia were pooled.

Parameters	G′	G″	G*	Phase Shift Angle	Complex Viscosity	Shear Strain %
Hardness	0.021	0.022	0.021	−0.182	0.021	−0.135
Springiness	−0.073	−0.096	−0.076	−0.198	−0.076	−0.104
Cohesiveness	0.174	0.138	0.170	−0.378	0.170	−0.340
Gumminess	0.037	0.022	0.035	−0.271	0.035	−0.205
Chewiness	0.030	0.010	0.028	−0.297	0.028	−0.204
Resilience	−0.046	−0.019	−0.044	0.125	−0.044	0.151

Correlation is significant at the 0.05 level (2-tailed); rheological parameters *n* = 13; textural parameters *n* = 23.

**Table 7 foods-12-03524-t007:** Pearson correlations between rheological and textural parameters of cookies. Data from WF and BSG-added cookies were pooled.

Parameters	G′	G″	G*	Phase Shift Angle	Complex Viscosity	Shear Strain %
Hardness	0.856 **	0.884 **	0.861 **	−0.324	0.862 **	−0.492
Fracturability	−0.364	−0.336	−0.363	0.007	−0.363	0.014
Springiness	0.063	0.025	0.058	−0.147	0.058	−0.144
Cohesiveness	0.416	0.325	0.409	−0.378	0.412	−0.223
Gumminess	0.896 **	0.889 **	0.899 **	−0.434	0.900 **	−0.546 *
Chewiness	0.471	0.424	0.466	−0.371	0.467	−0.396
Resilience	0.630 *	0.638 *	0.633 *	−0.232	0.636 *	−0.261

** Correlation is significant at the 0.01 level. * Correlation is significant at the 0.05 level (2-tailed); *n* = 15.

## Data Availability

The data used to support the findings of this study can be made available by the corresponding author upon request.

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
