# Peer review of "The Rheology and Textural Properties of Bakery Products Upcycling Brewers’ Spent Grain"

_foods, 2023, doi:10.3390/foods12193524_

Round 1

Reviewer 1 Report

The manuscript entitled “Rheology and textural properties of food products upcycling 2 brewers spent grain” deals with the addition of brewers' waste grain to baked products in order to give this residue value. The authors have shown that the addition of these grains with wheat flour to focaccia and cookies modified the characteristics of these products, but still, the products could be used as bakery products because they improved some qualities and it can also increase the nutritional value of the products. These results are interesting because it show the potential of reusing this material. The manuscript is well written and conclusions are supported by data.

Some revisions should be performed.

Abstract:

The abstract is well written but misses numbers. It is important to show at this stage how much it increased or decreased some parameters. It is important also to make it clear if those modifications in the products with the added grains are positive or negative for the product.

Introduction

Revise text

Revise Table 1 format. It is confusing.

Results

Table 1 legend: verify WF 20:20

English should be revised.

Author Response

REVIWER 1 Reply to Comments and Suggestions for Authors

Thank you to the reviewer for the precious suggestion.

Q1 The abstract is well written but misses numbers. It is important to show at this stage how much it increased or decreased some parameters. It is important also to make it clear if those modifications in the products with the added grains are positive or negative for the product.

R1: The percentage of increase, in comparison to the control have been added in the abstract

Q2Introduction

R2 The text in the Introduction has been revised

Q3 Revise Table 1 format. It is confusing.

R3. Table 1 has been revised to be more concise and clearer.

Q4 Results

Table 1 legend: verify WF 20:20

R4 We have revised the result section, and made a Result and Discussion section and the legends in all the tables fo consistency

Reviewer 2 Report

The topic of this manuscript is the use of brewers' stale (BSG) powder as a food ingredient to study its effect on the rheological properties of doughs and textural characteristics of baked products. The authors used different proportions of rye-based (RBSG) and barley-based (BBSG) brewers' stale powders in place of wheat flour to prepare simple model systems and complex food systems, which were characterized by a rheometer and a texture analyzer. The authors found that BSG addition increased the modulus and complex viscosity of the dough and decreased the flexibility and deformability of the dough. BSG addition also affected the textural characteristics such as hardness, elasticity, and mouthfeel of the baked products. Among them, RBSG-added Italian bread showed the highest hardness, chewiness, and stickiness, while BBSG-added cookies showed the highest fracture properties. There is room to optimize the framework and logic of this manuscript as well as the diagrams. Other questions were shown below:

1.    The introduction section could have been more comprehensive, including progress in research on the application of maltodextrin in other food products, such as cakes and pasta. Please refer this reference (Comprehensive Reviews in Food Science and Safety, 2023, Doi: 10.1111/1541-4337.13217).

2.    The theoretical basis for the experimental design could be expanded, such as the basis for the selection of the amount of malt dregs to be added.

3.    Reasons for choosing two food systems (focaccia and cookies) can be given in the introduction section. Please refer this reference (Journal of Food Processing and Preservation,2021, 45(9),e15684.).

4.    The typography of Table 1 is too crowded. Please adjust as appropriate to make it aesthetically pleasing.

5.    Some details can be added to the Materials and Methods section, such as physicochemical properties of the BSG powder such as particle size, water activity, and β-glucan content. The background should add the background of β-glucan (Critical Reviews in Food Science and Nutrition, 63(19): 3895-3911).

6.    The Materials and Methods section can also add structural parameters such as moisture content of dough and bakery products and combine these to assess food quality.

7.    The experimental part of the paper can be supplemented with the respective dough fermentation conditions in the model system, etc., so that others can repeat the experiments. The optimize experiment design can refer this reference(Journal of Agricultural and Food Chemistry, 2013, 61(23): 5526-5533).

8.    Figure 3 and Figure 4 can use the same range of axes to make the different samples more comparable.

9.    Note the use of conjunctions in the article, such assimple model- in line 541.

10. Sample labels such as WF1 in Figures 5 and 6 should be changed to match the presentation in the text. The coordinates can be expanded appropriately to show the full sample names.

The reference should be updated in recent years.

The topic of this manuscript is the use of brewers' stale (BSG) powder as a food ingredient to study its effect on the rheological properties of doughs and textural characteristics of baked products. The authors used different proportions of rye-based (RBSG) and barley-based (BBSG) brewers' stale powders in place of wheat flour to prepare simple model systems and complex food systems, which were characterized by a rheometer and a texture analyzer. The authors found that BSG addition increased the modulus and complex viscosity of the dough and decreased the flexibility and deformability of the dough. BSG addition also affected the textural characteristics such as hardness, elasticity, and mouthfeel of the baked products. Among them, RBSG-added Italian bread showed the highest hardness, chewiness, and stickiness, while BBSG-added cookies showed the highest fracture properties. There is room to optimize the framework and logic of this manuscript as well as the diagrams. This manuscript could be accepted after major revision. Other questions were shown below:

1.    The introduction section could have been more comprehensive, including progress in research on the application of maltodextrin in other food products, such as cakes and pasta. Please refer this reference (Comprehensive Reviews in Food Science and Safety, 2023, Doi: 10.1111/1541-4337.13217).

2.    The theoretical basis for the experimental design could be expanded, such as the basis for the selection of the amount of malt dregs to be added.

3.    Reasons for choosing two food systems (focaccia and cookies) can be given in the introduction section. Please refer this reference (Journal of Food Processing and Preservation,2021, 45(9),e15684.).

4.    The typography of Table 1 is too crowded. Please adjust as appropriate to make it aesthetically pleasing.

5.    Some details can be added to the Materials and Methods section, such as physicochemical properties of the BSG powder such as particle size, water activity, and β-glucan content. The background should add the background of β-glucan (Critical Reviews in Food Science and Nutrition, 63(19): 3895-3911).

6.    The Materials and Methods section can also add structural parameters such as moisture content of dough and bakery products and combine these to assess food quality.

7.    The experimental part of the paper can be supplemented with the respective dough fermentation conditions in the model system, etc., so that others can repeat the experiments. The optimize experiment design can refer this reference(Journal of Agricultural and Food Chemistry, 2013, 61(23): 5526-5533).

8.    Figure 3 and Figure 4 can use the same range of axes to make the different samples more comparable.

9.    Note the use of conjunctions in the article, such assimple model- in line 541.

10. Sample labels such as WF1 in Figures 5 and 6 should be changed to match the presentation in the text. The coordinates can be expanded appropriately to show the full sample names.

The reference should be updated in recent years.

Author Response

To REVIEWER 2 Reply to Comments and Suggestions for Authors

There is room to optimize the framework and logic of this manuscript as well as the diagrams. Other questions were shown below:

Thank you to the Reviewer for the precious suggestions, we have revised the manuscript to optimize framework, logic and diagrams and answered to the specific questions

Q1.The introduction section could have been more comprehensive, including progress in research on the application of maltodextrin in other food products, such as cakes and pasta. Please refer to this reference (Comprehensive Reviews in Food Science and Safety, 2023, Doi: 10.1111/1541-4337.13217).

R1 We found comprehensive interesting information in the suggested review that we though to summarize as In general, all food components (fiber, proteins, lipids, carbohydrates) participate in gel network generation, influencing the viscoelasticity of dough and enhancing the water binding capacity, as extensively reviewed in Wu et al [11].

Q2.The theoretical basis for the experimental design could be expanded, such as the basis for the selection of the amount of malt dregs to be added.

R2. We rephrase the description of the work objective (Line 78-82), in order to clarify the basis for our experimental design and added information in the result session (line 320-321)

Q3.Reasons for choosing two food systems (focaccia and cookies) can be given in the introduction section. Please refer this reference (Journal of Food Processing and Preservation,2021, 45(9),e15684.).

R3: With the aim to better clarify the reasons for choosing the focaccia and cookies food system we added in the introduction The two complex systems have been chosen as example for baking product largely consumed in the world for their nutrition and convenience (21). We also wanted to study the BSG behaviour in relation to yeast fermentation or to multiple ingredients to evaluate diverse upcycling possibility of BSG at 50%.

Q4.The typography of Table 1 is too crowded. Please adjust as appropriate to make it aesthetically pleasing.

R4: Thank you for the valuable comment, the table has been modified accordingly.

Q5. Some details can be added to the Materials and Methods section, such as physicochemical properties of the BSG powder such as particle size, water activity, and β-glucan content. The background should add the background of β-glucan (Critical Reviews in Food Science and Nutrition, 63(19): 3895-3911).

R5: We added some physicochemical properties of the BSG powder such as particle size, water activity, and β-glucan content in the BSG material description. In addition, some information from the available literature was added. These properties were also discussed under section 3.

Q6.The Materials and Methods section can also add structural parameters such as moisture content of dough and bakery products and combine these to assess food quality.

R6. We added moisture content of dough and bakery products in the result and discussion section 3.2.1 and 3.2.2 These properties were also discussed under section 3.

Q7.The experimental part of the paper can be supplemented with the respective dough fermentation conditions in the model system, etc., so that others can repeat the experiments. The optimize experiment design can refer this reference(Journal of Agricultural and Food Chemistry, 2013, 61(23): 5526-5533).

R7 The experimental part of the paper was described in detail in table 2 and in section 2.2, 2.3, and 2.4 . We added in this section some modification to have a cleared description of each step. No fermentation was carried out for the simple model systems but only for the focaccia food system described in section 2.3

Q8. Figure 3 and Figure 4 can use the same range of axes to make the different samples more comparable.

R.8: These figures were not made to compare focaccia with cookies since they are totally different products. Since we wanted to compare each BSG specific food with the related control (with no BSG addition) the axes scale were actually chosen in order to better visualize the differences within the same food model.

Q9. Note the use of conjunctions in the article, such as‘simple model-’ in line 541.

R9: Use of conjunction has been reviewed.

Q10.Sample labels such as WF1 in Figures 5 and 6 should be changed to match the presentation in the text. The coordinates can be expanded appropriately to show the full sample names.

R10: Figure 5 and 6 have been changed for a clearer and consistent description in agreement with the text.

Q11The reference should be updated in recent years.

R11 We have checked along the text and the last five years references were cited in most places.

Reviewer 3 Report

This paper presents the results of research into the rheological and textural properties of food products processed from brewer's spent grain.

However, there are still a number of ambiguities that need to be clarified. It is possible that some ambiguities arose from the structure of the language, so it would be good if the paper was read by a native English speaker.

Other comments refer to the presentation of the results, such as

table 1:

the text in the second column is overlyping with the text in the third column

figure 2:

observed are simple model systems or food sistemsy

it needs to be explained what present the letters and why are for some 

table 3: shear or shera (second and fourth column)

figures 3 & 4; some letters are missing that present signifficant similarity or differences 

figures 5b and 6b: red text is overlapping and to small, therefore, it is not possible to study the results

Sincerely

 It is possible that some ambiguities arose from the structure of the language, so it would be very usefull if the paper would read a native English speaker.

Author Response

To REVIEWER 3 Reply to Comments and Suggestions for Authors

Q1 This paper presents the results of research into the rheological and textural properties of food products processed from brewer's spent grain.

However, there are still a number of ambiguities that need to be clarified. It is possible that some ambiguities arose from the structure of the language, so it would be good if the paper was read by a native English speaker.

R1: Thank you to the reviewer for the precious comments, we have tried to improve the clarity revising some sentence all along the manuscript also with the help of a native English speaker.

Q2 table 1: the text in the second column is overlyping with the text in the third column

R2. Table 1 has been revised to be more concise and clearer.

Q3 figure 2:

observed are simple model systems or food sistemsy

it needs to be explained what present the letters and why are for some

R3 Figure 2 has been revised and more details added in the Figure caption.

Q4 table 3: shear or shera (second and fourth column)

R4 Typo has been corrected

Q5 figures 3 & 4; some letters are missing that present signifficant similarity or differences 

R5 Figure 3 and 4 have been revised and the missing letter have been added

Q6 figures 5b and 6b: red text is overlapping and to small, therefore, it is not possible to study the results

R6: Figure 5b and 6b have been changed and number added instead of the single variable to allow the study of the results. These number have been explained in each Figure caption

Reviewer 4 Report

The authors address an interesting topic, correlating rheological and textural properties. They could increase the quality of the manuscript, accepting brief observations highlighted in the attached document.

Author Response

To REVIEWER 4 Reply to Comments and Suggestions for Authors

Q1 The authors address an interesting topic, correlating rheological and textural properties.They could increase the quality of the manuscript, accepting brief observations highlighted in the attached document.

R1 Thank you for your comments, we have accepted your observations and applied some change to the text of the manuscript, as described below:

Q Line 364: All the BSG starch is hydrolyzed during the mashing process?

R Line364:Yes, since we run a quick test during the mashing process we can say that most starch was hydrolized, however since starch was not measured in our BSG flour we apply the following change the sentence : “should not apply to BSG flour since since most of it is expected to hydrolyze during the mashing process. Furthermore, the re-sidual quantity of starch BSG addition, should have a negligible influence when com-pared to the starch contribution of the wheat flour

Q Line440: sensory test results?

R Line440: in this sentence we wanted to compare sensory test results for hardness and chewiness from other studies to the instrumental texture analysis for hardness and chewiness. We understand that this may be confusing. Therefor we reduce and move the comments later in the discussion (Line 513-516) hoping to clarify this aspect: BSG addition acted positively on sensory quality such as hardness and crispiness for 20% BSG cookies baked at 200°C for 12 min [44]. Based on these data, the BSG added cookies are expected to give better quality scores in terms of sensory and further work must be done to ascertain this hypothesis.

Q Line 549 nutritional acceptability?

R Line 549 We revised the sentence now in Line 608-609: “nutritional value and textural acceptability.”

Reviewer 5 Report

In the title, the authors generalize with the word 'food,' but what type of food are they referring to? In their abstract, they mention that it is a dough supplemented with 50% of brewers spent grain. Please propose a title specific to their research topic and include the properties they are measuring in their study.

The abstract proposed by the authors is overly descriptive. They provide a clear description of the methodological process but fail to highlight any significant results; they merely describe them. They mention that their PCA analysis managed to discriminate or separate two groups; however, this needs to be clarified, and their abstract's conclusion is also unclear. In this regard, the authors should improve the abstract by promoting a version that emphasizes the use or importance of brewers spent grains and also discusses the impact on dough and baking compared to the use of common raw materials. One would expect to see the relevance of these data, which are entirely omitted in the current abstract.

Please review the journal template; the abstract should be limited to 200 words.

The introduction is strong and consistent with the topic the authors intend to publish. Additionally, the objective of the work is evident.

The authors have clearly described each of their processes in the results section, and the results are scientifically sound and robust. However, they are highly descriptive. They have omitted to conduct an in-depth discussion of the topic. When I read the manuscript, I expected that Section 4 would be the discussion section; however, Section 4 is the conclusions. In this regard, since the results presented by the authors are good, and they should not be moved since the authors have systematically conducted their research, I invite the authors to transform Section 4 into a discussion section. Here, they can compare each of their results with the existing literature. In this new results section, it seems crucial to compare with common raw materials, and the authors should propose the industrial benefit of using the 1:1 ratio they have suggested in different foods. This is essential to establish a precedent in their manuscript on the topic.

I consider that the conclusion section is more of a commentary that the authors have provided on their results. I suggest that this part be given a new format, relevant references be added, and it be integrated into the results section. In the conclusions, I would expect to see their key findings and what is novel about their work. They could even propose a research perspective. However, this is not present in this section, so a new conclusion should be proposed.

I believe the manuscript is good and aligns with the scope of the journal. However, there are still significant modifications that need to be made to it. Therefore, I recommend that major revisions be provided for this manuscript, and if the authors consider and address the comments, they should have the opportunity to resubmit it for reevaluation.

Author Response

To REVIEWER 5 Reply to Comments and Suggestions for Authors

In the title, the authors generalize with the word 'food,' but what type of food are they referring to? In their abstract, they mention that it is a dough supplemented with 50% of brewers spent grain. Please propose a title specific to their research topic and include the properties they are measuring in their study.

Response: Thank you for the precius suggestion. We are aiming to give a title more focused on the upcycling viability of BSG. However, the type of food is given in the title now.

The abstract proposed by the authors is overly descriptive. They provide a clear description of the methodological process but fail to highlight any significant results; they merely describe them. They mention that their PCA analysis managed to discriminate or separate two groups; however, this needs to be clarified, and their abstract's conclusion is also unclear. In this regard, the authors should improve the abstract by promoting a version that emphasizes the use or importance of brewers spent grains and also discusses the impact on dough and baking compared to the use of common raw materials. One would expect to see the relevance of these data, which are entirely omitted in the current abstract.

Response: Thank you for the precious comments: the abstract was revised and significant results with industrial relevance have been added.

Please review the journal template; the abstract should be limited to 200 words.

Response: Thanks for your comments, we revised the text as per the comment.

The introduction is strong and consistent with the topic the authors intend to publish. Additionally, the objective of the work is evident.

Response: Thanks for your comments.

The authors have clearly described each of their processes in the results section, and the results are scientifically sound and robust. However, they are highly descriptive. They have omitted to conduct an in-depth discussion of the topic. When I read the manuscript, I expected that Section 4 would be the discussion section; however, Section 4 is the conclusions. In this regard, since the results presented by the authors are good, and they should not be moved since the authors have systematically conducted their research, I invite the authors to transform Section 4 into a discussion section. Here, they can compare each of their results with the existing literature. In this new results section, it seems crucial to compare with common raw materials, and the authors should propose the industrial benefit of using the 1:1 ratio they have suggested in different foods. This is essential to establish a precedent in their manuscript on the topic.

Response: thank you for your precious observation. We have actually forgot to add the word “discussion” to section 3. In fact, since the journal allow to choose for the comprehensive section, we have preferred to discuss the results after their presentation, in section 3, compared our results with the existing literature, also adding some more finding from other studies. All the reference have been highlighted in red. Changes have been highlighted in red.

I consider that the conclusion section is more of a commentary that the authors have provided on their results. I suggest that this part be given a new format, relevant references be added, and it be integrated into the results section. In the conclusions, I would expect to see their key findings and what is novel about their work. They could even propose a research perspective. However, this is not present in this section, so a new conclusion should be proposed.

Response: The conclusion part is a separate section 4 (while the refence were left for the discussion in section 3) . As per your comments, the industrial benefits of upcycling 1:1 ratio are added now with a few more discussions related to our study. The keys outcome were highlighted in red in the conclusion part, please refer lines 598 to 601 and 612 to 614.

I believe the manuscript is good and aligns with the scope of the journal. However, there are still significant modifications that need to be made to it. Therefore, I recommend that major revisions be provided for this manuscript, and if the authors consider and address the comments, they should have the opportunity to resubmit it for reevaluation.

Response: Thank you again for your valuable comments and consideration.

Round 2

Reviewer 2 Report

The author has responded to the reviewer's comment point by point. However, there are still some issues that need to be clarified. The reference should be updated in recent years according to the requirement of the Journal with Doi. 

The author has responded to the reviewer's comment point by point. However, there are still some issues that need to be clarified. The reference should be updated in recent years according to the requirement of the Journal with Doi.